# Response to comment on 'Naked mole-rat mortality rates defy Gompertzian laws by not increasing with age'

J Graham Ruby, Megan Smith, Rochelle Buffenstein*

Calico Life Sciences, LLC, South San Francisco, United States

**Abstract** For most adult mammals, the risk of death increases exponentially with age, an observation originally described for humans by Benjamin Gompertz. We recently performed a Kaplan–Meier survival analysis of naked mole-rats (*Heterocephalus glaber*) and concluded that their risk of death remains constant as they grow older (Ruby et al., 2018). Dammann et al. suggest incomplete historical records potentially confounded our demographic analysis (Dammann et al., 2019). In response, we applied the left-censorship technique of Kaplan and Meier to exclude all data from the historical era in which they speculate the records to be confounded. Our new analysis produced indistinguishable results from what we had previously published, and thus strongly reinforced our original conclusions.

DOI: https://doi.org/10.7554/eLife.47047.001

## Introduction

Dammann et al. express concern that only a small fraction of the naked mole-rat lifespan (4400 days, or ~12 years) has been used to conclude that hazard rates are constant (*Dammann et al., 2019*). This concern derives from the preternaturally long lifespans experienced by an apparently non-aging mammal in a relatively safe, captive environment: at ages far beyond their allometrically-expected lifespans, or their lifespans in the wild, only a minority of the naked mole-rat population has expired. Unlike for organisms with Gompertzian hazard, the concept of a maximal lifespan is impossible to define meaningfully for a non-aging species, a conundrum that was discussed as follows in our original paper: "Under an exponential decay model, the concept of a 'maximal lifespan' loses relevance: at no point does mortality hazard grow into an insurmountable obstacle. One could try and force the concept of 'maximal lifespan' onto an exponentially-decaying species by defining it in terms of a high percentile of lifespan (say, 95%) – and by that definition, the maximal lifespan of the naked mole-rat would be very long indeed! But that definition ignores the implicit meaning of the term: as an age beyond which biological wear and damage has become insurmountable" (*Ruby et al., 2018*).

The question remains: how long must a lack of aging be observed in order to be considered long-enough? Dammann et al. remain attached to the notion of maximal lifespan, despite the logical conundrum discussed above. In contrast, we considered the span of time across which our conclusions were irrefutable (4400 days, or ~12 years) in the contexts of their allometrically-expected full lifespans, or their lifespans as measured in the wild: "That twelve year stretch of time can be considered from several perspectives. In Figure 5, it is considered from the perspective of $T_{sex}$, that is the minimum lifespan allowable for an organism to reproduce. Most mammals live beyond that age, as is required to provide progeny with support and nurturing (*Lee, 2003*), but show signs of demographic aging within a few fold of $T_{sex}$; from this perspective, the naked mole-rat is exceptional, showing no signs of demographic aging many dozens-of-fold beyond $T_{sex}$ (Figure 5). Twelve years can also be considered in the context of the expected total lifespan of a naked mole-rat based on its body size (*de Magalhães et al., 2007*): this 35-gram rodent would be expected to live up to six

*For correspondence:
rbuffen@calicolabs.com

years, but instead has not shown the first sign of demographic aging at twice that age. Twelve years could finally be considered from the perspective of natural lifespan in the wild, which is estimated to be 2–3 years for the naked mole-rat (*Hochberg et al., 2016*)" (*Ruby et al., 2018*).

The argument in *Ruby et al. (2018)* relies on observations within the first 12 years of life, which means that the objections of Dammann et al. to the sizes of error bars at older ages are not relevant (though we stand behind our decision to communicate the full scope of available data). But even if one is not satisfied by our arguments, we seriously considered these concerns in our original manuscript, discussing them at length: "We cannot reject the possibility of an as-yet-undetectable Gompertzian component to naked mole-rat mortality. And in fact, this alternative model is not formally falsifiable: any observational extension of constant hazard would simply delay the earliest point at which Gompertzian hazard might take effect. However, given the extremely low baseline hazard rate of 1/10,000 per day for non-breeders and ~1/100,000 for breeders, it seems unlikely that the Gompertzian hazard is being overwhelmed. But even under a delayed-aging model, naked mole-rats profoundly distinguish themselves from other mammals: in this context, through an unprecedentedly-long adulthood prior to the emergence of Gompertzian hazard" (*Ruby et al., 2018*).

## Results and discussion

### Bias due to missing death records

Dammann et al. express the concern that "the data set could be biased due to missing death records before 2008" (*Dammann et al., 2019*). To address this concern, we have re-analyzed our published data including only observations made on or after January 1, 2008. For animals born prior to that date, observational data (of them being alive) was truncated using left-censorship (that is, truncation on the left). This approach is recommended in the original publication paper by Kaplan and Meier, where it is deemed appropriate "if it is desired to permit the entrance of items [i.e. animals] into the sample after the commencement of their lifetimes" (page 463 of *Kaplan and Meier, 1958*). Such is the case here. Dammann et al. assert that "methods for addressing left censoring require information on the number of missing animals". To the contrary, Kaplan and Meier explicitly describe the lack of such information as an assumption of their technique: "It is assumed that nothing is known of the existence of any such item [i.e. animal] that dies before it becomes available for observation" (*Kaplan and Meier, 1958*). Since Dammann et al. do not question the validity of data collected in or after 2008, left truncation at January 2008 should address their concern of alleged bias.

We applied this standard technique, and *Figure 1A* shows juxtaposed Kaplan–Meier survival analysis with left-censorship at January 1, 2008 (orange) along with our original Kaplan–Meier analysis (Figure 1 of *Ruby et al., 2018*; green). The survival plot was minimally affected by this new analytical framework, with final survival still above 50% (ending at 51.9% rather than 61.6%). The new analysis continued to match the expected trajectory for constant daily mortality hazard of $8 \times 10^{-5}$ (*Figure 1A*; dashed purple line). The differences between our original and new analyses were therefore insufficient to warrant any modification to our original conclusions about the median lifespan for $T_{sex}$-surviving naked mole-rats being ~6900 days (19 years).

We used the left-censored Kaplan–Meier survival curve to calculate the age-specific mortality hazard across nine age bins (*Figure 1B*, orange). Those new estimates failed to show an increase with age and were statistically indistinct versus the age-specific hazards presented in our original manuscript (*Figure 1B*, green). We therefore maintain our original conclusion: that naked mole-rat mortality hazard does not increase with age, with that assertion being statistically robust to the beginning of our final estimate bin (6,529 days, or 18 years).

### Conclusions

The central finding of *Ruby et al. (2018)* – that naked mole-rats defy Gompertzian mortality laws – was based on Kaplan–Meier survival and hazard analyses on all the high-quality, full-lifespan data at our disposal. We followed that up with analyses inclusive of low-quality (imprecise) data (Figure 2 of *Ruby et al., 2018*), as well as independent analyses of each sex and identifiable social class (i.e.

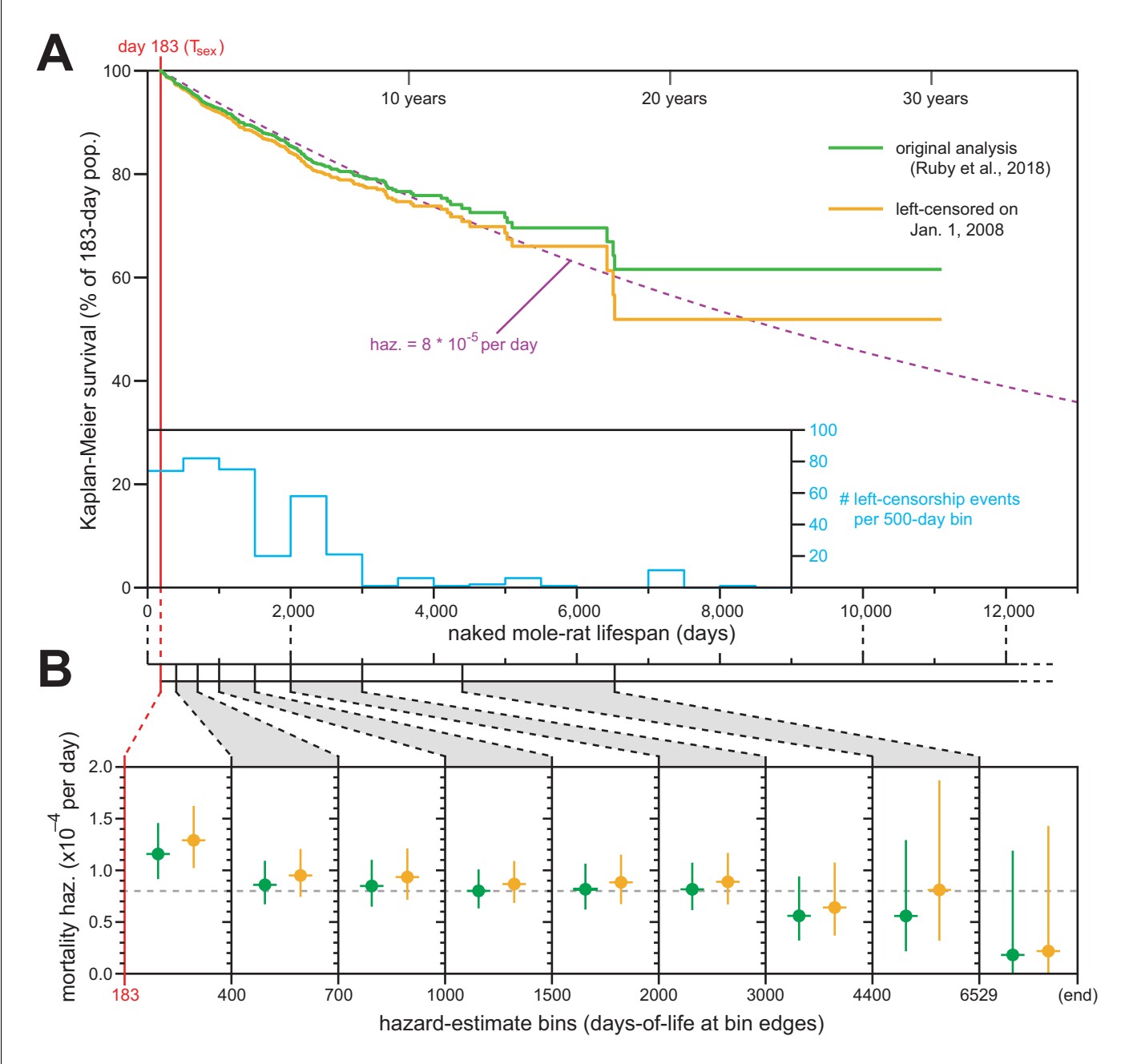

**Figure 1.** Exclusion of all pre-2008 lifespan data through left-censorship did not modify the observed lifespan demographics of *H. glaber*. (**A**) Kaplan–Meier survival curve for naked mole-rats after reaching reproductive maturity ($T_{sex}$; 6 months from birth; 183 days; red). Green: original calculation (Figure 1 of *Ruby et al., 2018*). Orange: calculation on the same data, left-truncated on January 1, 2008 (see Materials and methods). Purple: expected survivals from $T_{sex}$ given a constant mortality hazard of $8 \times 10^{-5}$ per day (*Ruby et al., 2018*). Inset: a histogram of left-censorship events. (**B**) Hazard estimates across each of the lifespan bins (from Figure 1B of *Ruby et al., 2018*), calculated for both of the survival plots from panel (**A**), colored as in panel (**A**). Vertical bars indicate 95% confidence intervals. Dotted grey line indicates $8 \times 10^{-5}$ deaths per day.
DOI: https://doi.org/10.7554/eLife.47047.002

breeders, non-breeders) of naked mole-rats (Figures 3 and 4 of *Ruby et al., 2018*). Each of those analyses reinforced our original conclusion. Further, to address the concern of Dammann et al. that our records from pre-2008 could potentially have introduced bias to our analyses, we performed a

new analysis. In light of the robust agreement between our original analysis (*Ruby et al., 2018*) and our new analysis, we stand by all of the conclusions in *Ruby et al. (2018)*.

## Materials and methods

Kaplan–Meier survival analysis was performed on the data from Supplementary file 1 from *Ruby et al. (2018)*, filtered as described for Figure 1 of that paper (the 'initial analysis': animals with day-resolution birth and death/censorship data), again according to the method of *Kaplan and Meier (1958)*, but now truncating lifespan observations on the left at January 1, 2008. For example: an animal *A* born on after January 1, 2008 was considered identically as in *Ruby et al. (2018)*. In contrast, an animal *B* born one year prior (on January 1, 2007), would not have been counted as a member of the population until day 365. For that animal, as expressed by equations 2a and 2b of *Kaplan and Meier (1958)*, one count representing that animal would be added to $n_j$ at $j$ = 365.

Age-specific per-day hazard was calculated across bins (as described in *Ruby et al., 2018*), with left-censorship again taken into account. Using the examples above: animal *A* would be considered identically to the original analysis. For the first bin (days 183 through 399), if animal *A* survived that bin, then it would have contributed zero deaths to the numerator and 217 days to the denominator of that bin (i.e. the length of the bin); had it died at 380 days, then it would have contributed one death to the numerator and 198 days to the denominator (the number of days it was observed to be alive in that bin, plus the day it died). In contrast, if animal *B* had survived that bin, it would have contributed only 35 days to the denominator (its entry point at day 365 through day 399). Had animal *B* died at 380 days, then it would have contributed one death to the numerator and only 16 days to the denominator (the number of days it was observed alive post-truncation, plus the day it died).

## Additional information

### Competing interests

J Graham Ruby, Megan Smith, Rochelle Buffenstein: This research was funded by Calico Life Sciences LLC, where all authors were employees at the time the study was conducted. The authors declare no other competing financial interests.

### Funding

| Funder | Author |
| --- | --- |
| Calico Life Sciences, LLC | Rochelle Buffenstein |

The authors were all employees of Calico Life Sciences, LLC at the time the study was conducted.

### Author contributions

J Graham Ruby, Conceptualization, Data curation, Formal analysis, Visualization, Methodology, Writing—original draft, Writing—review and editing; Megan Smith, Conceptualization, Data curation, Writing—original draft; Rochelle Buffenstein, Conceptualization, Data curation, Writing—original draft, Project administration, Writing—review and editing

### Author ORCIDs

J Graham Ruby (ID) https://orcid.org/0000-0002-4798-5420
Rochelle Buffenstein (ID) https://orcid.org/0000-0003-3285-8311

### Decision letter and Author response

Decision letter https://doi.org/10.7554/eLife.47047.005
Author response https://doi.org/10.7554/eLife.47047.006

## Additional files

### Data availability

All data used in this paper were published as part of Ruby et al., 2018.

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
