## [Decision Letter]

Thank you for submitting your article 'Response to comment on 'Naked mole-rat mortality rates defy Gompertzian laws by not increasing with age' to *eLife*. Your article has been reviewed by two peer reviewers, and the evaluation has been overseen by a Reviewing Editor (Michael Rose), a Senior Editor (Patricia Wittkopp), and the *eLife* Features Editor (Peter Rodgers). The following individuals involved in review of your submission have agreed to reveal their identity: Laurence Mueller (Reviewer #1); Caleb Finch (Reviewer #2).

The reviewers supported publication of your article subject to the following point being addressed in a revised version.

Essential revisions:

1) Dammann et al. raise the concern that only a small fraction of the lifespan has been used to conclude hazard rates are constant. Even with this small range, the confidence interval on hazard rates in the last age range are quite large. The authors of this Response should address this concern.

---

## [Author Response]

Essential revisions:1) Dammann et al. raise the concern that only a small fraction of the lifespan has been used to conclude hazard rates are constant. Even with this small range, the confidence interval on hazard rates in the last age range are quite large. The authors of this Response should address this concern.

We once again address this discussion point in our revised manuscript and highlight where in our original paper we had originally discussed the concerns that have been re-raised by Dammann et al. in their comment. We have added substantial text addressing this concern, largely by restating points made in the Discussion section of the original paper (Ruby et al., 2018). As before, our re-analyses of the data using left censorship (submitted in the 1st draft of the commentary) strongly support our original conclusions.